# Supramolecular Kandinsky circles with high antibacterial activity

Heng Wang[1], Xiaomin Qian[1,2], Kun Wang [3], Ma Su[1], Wei-Wei Haoyang[4], Xin Jiang[5], Robert Brzozowski[6], Ming Wang[5], Xiang Gao[7], Yiming Li[1], Bingqian Xu[3], Prahathees Eswara[6], Xin-Qi Hao[7], Weitao Gong [2], Jun-Li Hou[4], Jianfeng Cai[1] & Xiaopeng Li[1]

Nested concentric structures widely exist in nature and designed systems with circles, polygons, polyhedra, and spheres sharing the same center or axis. It still remains challenging to construct discrete nested architecture at (supra)molecular level. Herein, three generations (**G2**−**G4**) of giant nested supramolecules, or Kandinsky circles, have been designed and assembled with molecular weight 17,964, 27,713 and 38,352 Da, respectively. In the ligand preparation, consecutive condensation between precursors with primary amines and pyrylium salts is applied to modularize the synthesis. These discrete nested supramolecules are prone to assemble into tubular nanostructures through hierarchical self-assembly. Furthermore, nested supramolecules display high antimicrobial activity against Gram-positive pathogen methicillin-resistant *Staphylococcus aureus* (MRSA), and negligible toxicity to eukaryotic cells, while the corresponding ligands do not show potent antimicrobial activity.

[1] Department of Chemistry, University of South Florida, Tampa, FL 33620, USA. [2] State Key Laboratory of Fine Chemicals, School of Chemical Engineering, Dalian University of Technology, Dalian, Liaoning 116024, China. [3] Single Molecule Study Laboratory, College of Engineering and Nanoscale Science and Engineering Center, University of Georgia, Athens, GA 30602, USA. [4] Department of Chemistry, Fudan University, Shanghai 200433, China. [5] State Key Laboratory of Supramolecular Structure and Materials, College of Chemistry, Jilin University, Changchun, Jilin 130012, China. [6] Department of Cell Biology, Microbiology and Molecular Biology, University of South Florida, Tampa, FL 33620, USA. [7] College of Chemistry and Molecular Engineering, Zhengzhou University, Zhengzhou, Henan 450001, China. These authors contributed equally: Heng Wang, Xiaomin Qian  Correspondence and requests for materials should be addressed to W.G. (email: wtgong@dlut.edu.cn) or to J.-L.H. (email: houjl@fudan.edu.cn) or to J.C. (email: jianfengcai@usf.edu) or to X.L. (email: xiaopengli1@usf.edu)

Nested concentric patterns are common throughout nature and designed systems within broad scales. Examples of nested structures in nature include onion, tree-rings, spider webs, Russian dolls, circular surface waves on the water, concentric hexagons at Saturn's North Pole[1], and so forth. In the art field, concentric rings are also known as Kandinsky circles named after Wassily Kandinsky, a pioneer in abstract art, because of his prominent and profound painting Color Study; Squares with Concentric Circles[2]. In addition to art and mathematics study[3], chemists have been fascinated by nested architectures with discovery and/or creation of a significant number of two-dimensional (2D) and three-dimensional (3D) systems that display nested layer and shell arrangements of atoms, molecules, and materials. This is evident in the, for example, structure of Par-iacoto virus[4], light harvesting complexes in purple bacteria[5],

atomic organization of carbon onions[6], concentric arrangement of multi-walled carbon nanotubes[7], self-assembly of amphiphilic dendrimers[8], block-copolymers[9], and graphene onion rings[10]. It still remains, however, a formidable challenge to construct discrete nested structures at (supra)molecular level. Up to date, only a very few discrete nested supramolecules were reported but limited to DNA nanostructures[11,12], porphyrin nanorings[13], and complexes of carbon nanorings with fullerenes[14,15]. Nonetheless, precise control over the self-assembly toward discrete supramolecular nested structures is still one of the ultimate goals and challenges in the field.

Inspired by Kandinsky circles, we focused our attention on seeking of an efficient (supra)molecular expression of the nested structure based on coordination-driven self-assembly. Since early 1990s[16], coordination-driven self-assembly has witnessed an

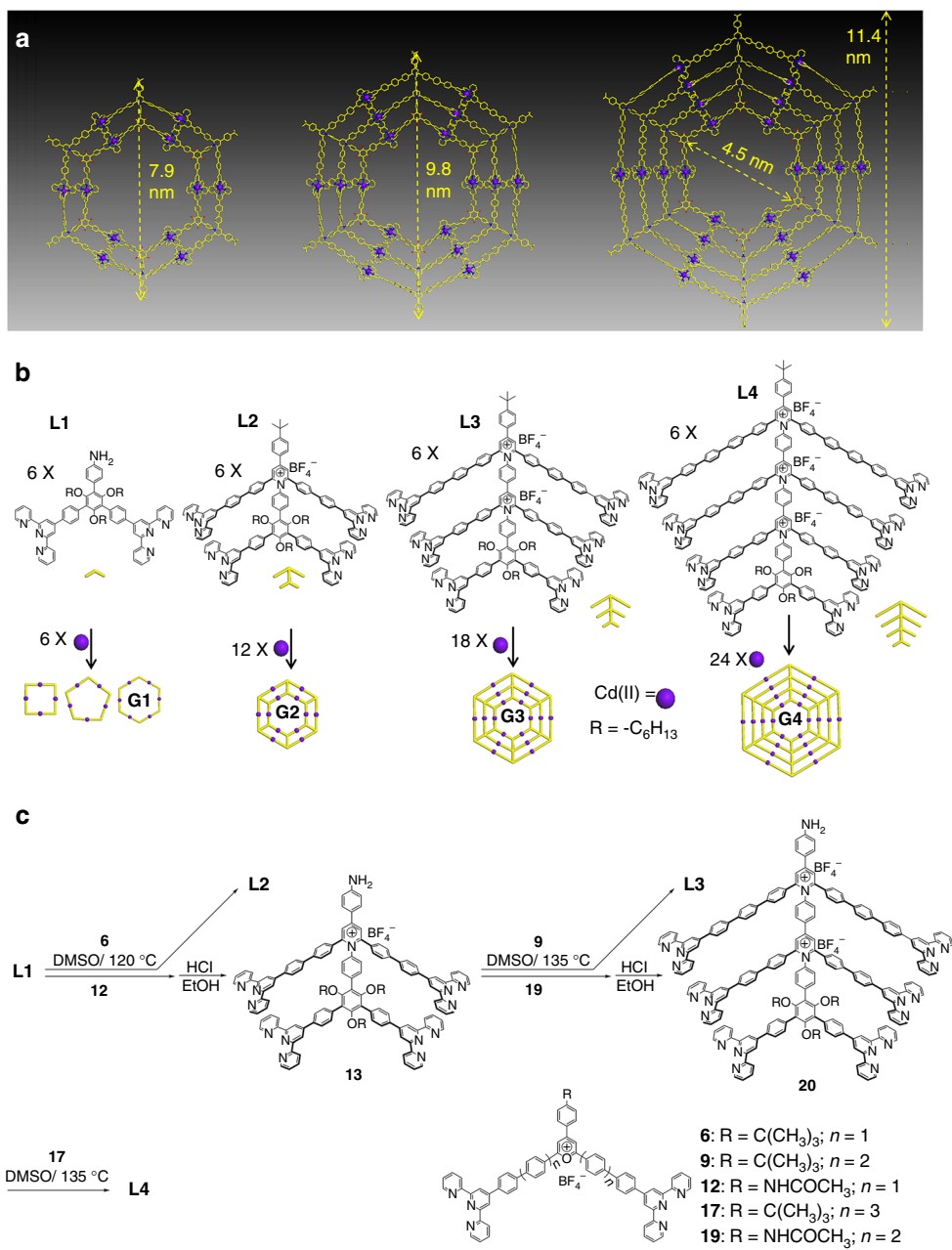

**Fig. 1** Synthesis and self-assembly of nested hexagon. **a** Molecular modeling of nested hexagons (**G2**–**G4**). Alkyl chains were omitted for clarity in the molecular models. **b** Self-assembly of **G1**–**G4**. In the self-assembly of **L1**, a mixture of macrocycles was obtained instead of discrete hexagon. **c** Synthesis of the ligands **L1**–**L4** based on pyrylium and pyridinium salts chemistry

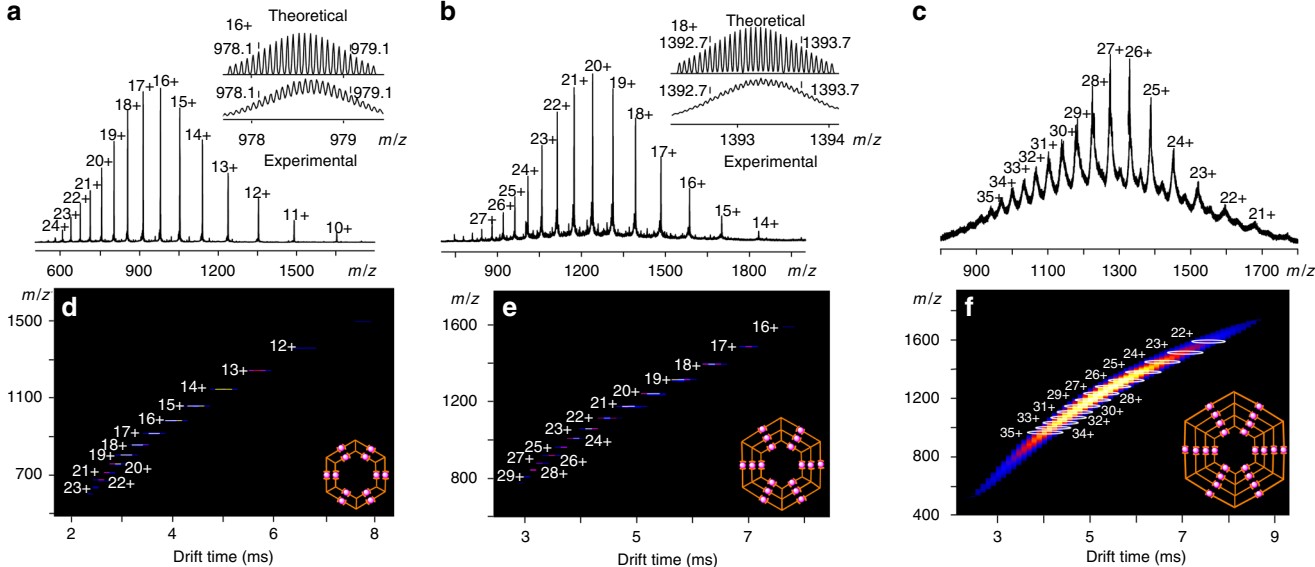

**Fig. 2** ESI-MS and TWIM-MS of nested hexagons. **a** ESI-MS of **G2**, **b** ESI-MS of **G3**, **c** ESI-MS of **G4**, **d** TWIM-MS of **G2, e** TWIM-MS of **G3**, and **f** TWIM-MS of **G4**

explosion in constructing various 2D and 3D supramolecular architectures[17−25] owing to the highly directional and predictable feature of coordination, as well as the structural information instilled in the building blocks. Benefiting from their precisely controlled structures, such supramolecules have found a myriad of applications[26−33]. With nested hexagons or Kandinsky circles as our target (Fig. 1a, b), we should be able to create a series of discrete supramolecules, e.g., generations 1 to 4 (**G1**−**G4**) in the self-assembly of multitopic organic ligands with metal ions as bridging units. However, such design posed a substantial challenge in the synthesis of multitopic terpyridine (tpy) ligands given the increasing size and complexity of the target. In this study, we overcome the challenge through using modular pyrylium salts followed by consecutive condensation reaction with primary amines to prepare multitopic pyridinium salts ligands. As such, we successfully assemble giant discrete nested hexagons from **G2** to **G4**.

In addition to self-assembly of discrete nested hexagons, these giant multilayered supramolecules have a strong tendency to form tubular nanostructures through hierarchical self-assembly. We envision that these nested hexagons could interact with bacterial lipid membrane and form transmembrane channels owing to their desired shapes and sizes, cationic scaffolds with precise charge positioning, delicate balance of the overall hydrophobicity/hydrophilicity, and remarkable rigidity and stability. Indeed, these nested hexagons display highly potent antimicrobial activity against methicillin-resistant *Staphylococcus aureus* (MRSA) and negligible toxicity to eukaryotic cells. It is worth noting that few synthetic ion channels based on metal-organic polyhydra and prism are reported, but none of them are active against bacteria[34,35]. Here we conduct a detailed study to address this antimicrobial mechanism, hopefully to shed light on fighting bacterial infections and antibiotic resistance.

## Results
**Synthesis and hierarchical assembly of supramolecular Kandinsky circles G2−G4.** Based on geometry and topology analysis along with detailed molecular dynamic simulation, we reasoned a series of nested hexagons could be assembled, e.g., **G2**−**G4** or even higher generation. In addition to the seeking of the chemical

expression of the beautiful nested geometry, such architectures with increasing multivalent interaction[36] or high density of coordination sites[37] would advance the complexity and stability of 2D supramolecules. Moreover, the large and rigid scaffolds with all conjugated backbones may facilitate further packing of nested hexagons into tubular-like nanostructures through hierarchical self-assembly[38]. The major challenge posed in such design is the synthesis of multitopic ligands. This motivated us to exploit the synthesis of multitopic tpy ligand with the goal of developing a robust synthetic approach. We initially employed a synthetic strategy based on Suzuki coupling reaction on the dibromo-pyridinium salts as shown in Supplementary Figure 1 according to our recent study[39]. The synthesis and separation of singly charged tetratopic tpy ligand **L2** was readily accomplished with good yield. In the preparation of **L3**; however, it was extremely difficult to isolate the doubly charged hexatopic ligand from the system, which contained multi-armed byproducts, e.g., pentatopic tpy component with very similar polarity and size as target ligand. Similar isolation challenge also obstructed the preparation of **L4**.

After numerous unsuccessful attempts with several separation techniques, we revised the strategy and developed an approach to modularize the synstheis as shown in Fig. 1c. Briefly, we first synthesized a series of ditopic tpy pyrylium salts (**6**, **9**, **12**, **17**, and **19**) as modules for condensation via performing Suzuki coupling reaction on dibromo-pyrylium salts (Supplementary Figure 2a±d). At the basic conditions, dibromo-pyrylium salts underwent ring opening to form neutral diketones species[40], which could be easily isolated using regular alumina column chromatography. After that, ditopic tpy pyrylium salts could be quantitatively obtained through ring-closing with strong acid, e.g., $HBF_4$ treatment. All of these ditopic tpy pyrylium salts were subjected for the subsequent condensation reactions with the precursors with primary amines to generate pyridinium salts ligands or intermediates. Combined with sequential condensation and deprotection of the precursors with primary amines, this simple but powerful strategy allowed us to prepare multitopic tpy ligands, i.e., **L2**−**L4**, perhaps even higher generation of multitopic building blocks if needed. It is also worth noting that regular column separation was readily achievable because of vast polarity and size difference determined by numbers of charge and tpy

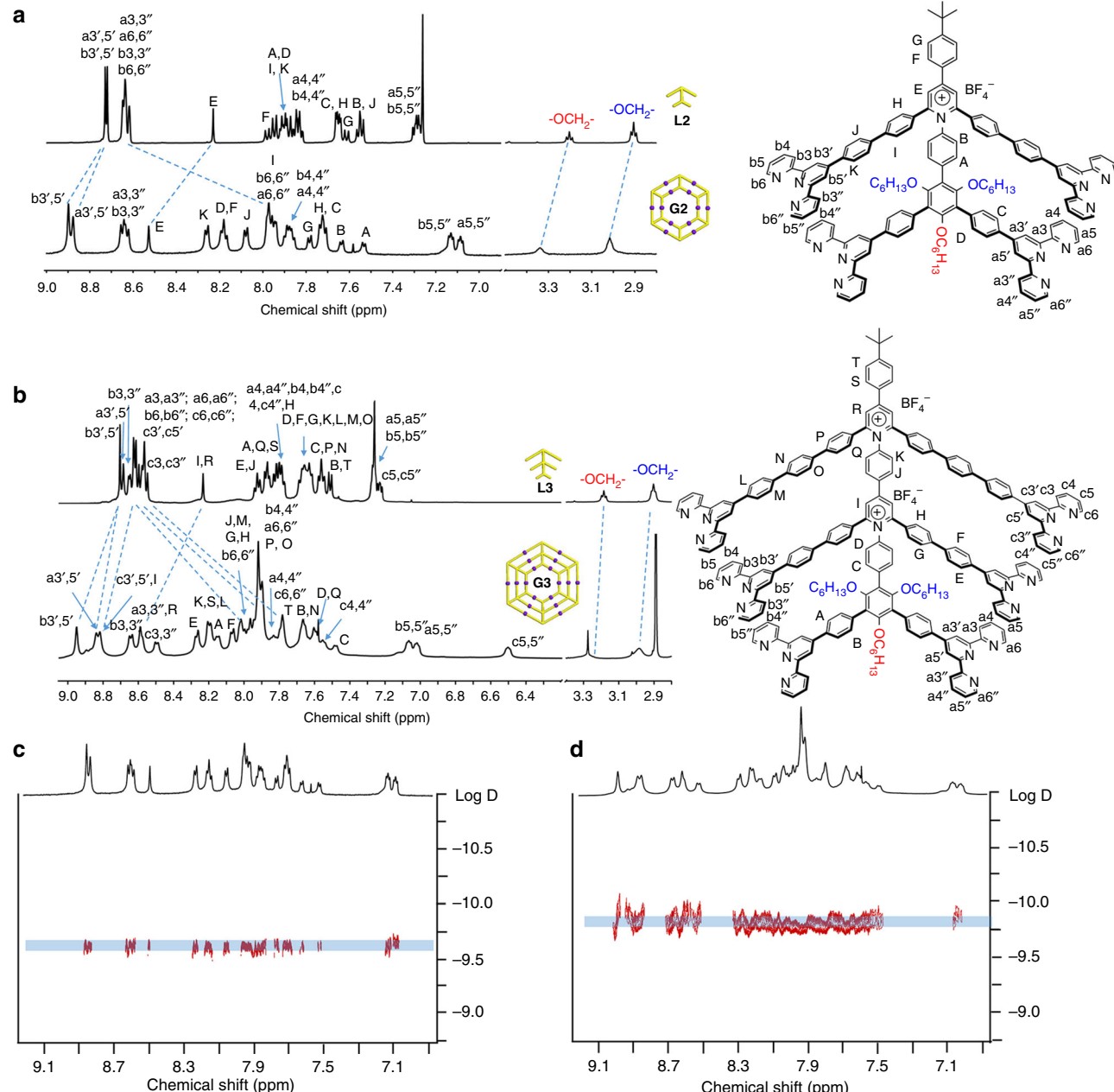

**Fig. 3** NMR study of nested hexagons. **a** [1]H NMR spectra of **L2** in CDCl$_3$ and **G2** in CD$_3$CN. **b** [1]H NMR spectra of **L3** in CDCl$_3$ and **G3** in CD$_3$CN. 2D DOSY spectra of **c G2** and **d G3**

between amino-precursors and products. All of the ligands and intermediates were fully characterized by [1]H, [13]C, 2D correlation spectroscopy (2D-COSY), and/or nuclear Overhauser effect spectroscopy (NOESY) nuclear magnetic resonance (NMR) as well as high-resolution electrospray ionization time-of-flight mass spectrometry (ESI-TOF-MS) (Supplementary Figures 3–85). Note that each **L4** was observed with strong binding affinity to two water molecules as evidenced by the ESI-TOF-MS obtained by regular solvent and dry solvent (Supplementary Figures 84 and 85). We speculated that the triply charged scaffold of pyridinium salts **L4** might have strong interaction with water molecules.

With these multitopic ligands in hand, we next carried out self-assembly with Cd(II) rather than Zn(II) in our previous study[39] because of high reversibility of Cd(II) to ensure formation of discrete giant nested hexagons in high yields. In self-assembly, multitopic ligands **L2−L4** were mixed with Cd(NO$_3$)$_2$ at ratio 1:2,

1:3, and 1:4, respectively. After 3 h of incubation at 50 °C, the assemblies were precipitated via adding excessive NH$_4$PF$_6$. After a simple wash with water, supramolecules were obtained and directly subjected for characterization without further purification. ESI-MS was first used to address the molecular composition of three nested hexagons (Fig. 2a–c). As expected, discrete hexamers were identified as the target assemblies with measured molecular weight at 17,964, 27,713, and 38,352 Da for **G2**, **G3**, and **G4**, respectively. The experimental isotopic patterns of each charge state of **G2** and **G3** by losing PF$_6$⁻ are in good agreement with simulated isotopic distributions (Supplementary Figures 115 and 116). Due to the high molecular weight of **G4** beyond resolution limits of ESI-TOF MS, we were not able to obtain high-resolution isotope patterns for each charge state. Traveling wave ion mobility-mass spectrometry (TWIM-MS)[37] was further employed to address the shapes and sizes of nested hexagons

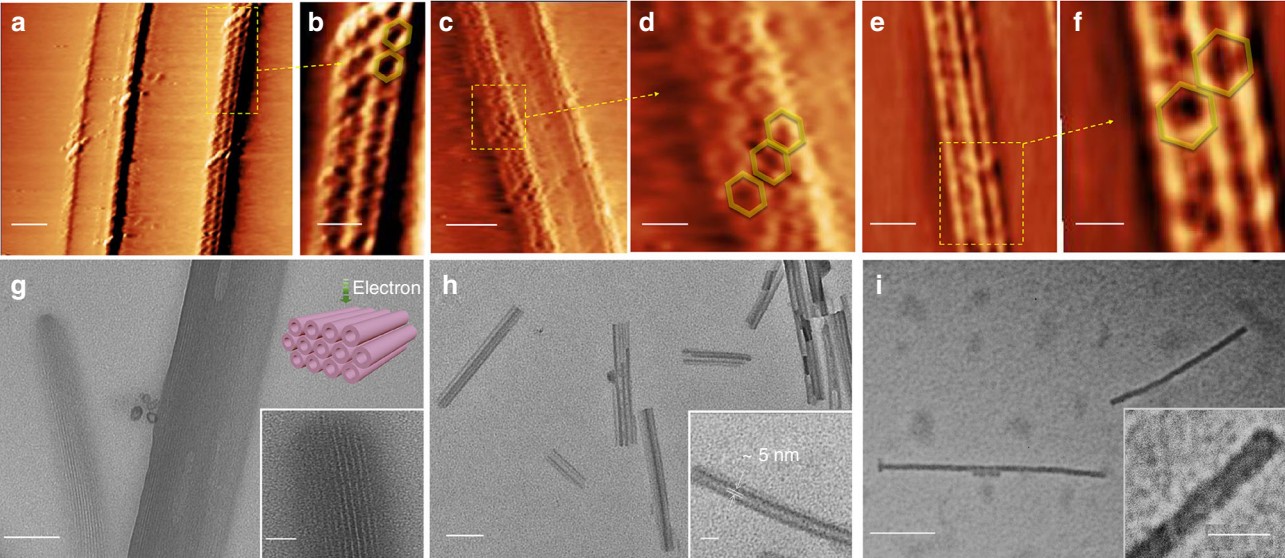

**Fig. 4** Imaging of nested hexagons and tubular-like nanostructures assembled by nested hexagons. STM images of nanoribbons assembled by **a, b G2** (scale bar, 25 and 10 nm, respectively); **c, d G3** (scale bar, 25 and 10 nm, respectively); and **e, f G4** (scale bar, 25 and 10 nm, respectively) on HOPG surface. TEM imaging of tubular-like nanostructures assembled by **g G2** (scale bar, 100 and 20 nm for zoom-in image); **h G3** (scale bar, 100 and 20 nm for zoom-in image); and **i G4** (scale bar, 100 and 20 nm for zoom-in image)

(Fig. 2d–f). The narrow drifting time band at each charge state indicated no other isomers or conformers existed. These experimental collision-cross sections (CCSs) of shape-persistent hexagons were calculated as 2466, 3660, and 5009 $Å^2$, agreeing well with theoretical CCSs at 2547, 3728, and 5002 $Å$ for **G2−G4**, respectively (Supplementary Table 1, Supplementary Figures 118–120). Finally, gradient tandem mass spectrometry (gMS$^2$) [37] were employed to compare the stability of these nested structures, which were completely dissociated at 14, 24, and 31 V, respectivly (Supplementary Figure 117a–c), suggesting a substantial increase of stability in addition to their increasing complexity. In contrast, the self-assembly of **L1** generated a mixture of macrocycles instead of discrete hexagons (Supplementary Figure 114).

$^1$H NMR of the complexes **G2−G3** together with their corresponding ligands are shown in Fig. 3. The spectrum of complex **G2** showed two sets of tpy group signals (Fig. 3a), suggesting the formation of a highly symmetric structure. In comparison with **L2**, the characteristic upfield shift for the 6,6″-tpy protons ($\Delta\delta \approx 0.65$ ppm) was observed after the complexation reaction. By increasing layers of nested hexagons, similar upfield shifts are also distinguished ($\Delta\delta \approx 0.70$ ppm **G3/L3**, $\Delta\delta \approx 0.60$ ppm **G4/L4**, Fig. 3b, Supplementary Figure 110), although more complicated resonance patterns were displayed in the spectra, due to the overlap of large numbers of phenyl and tpy groups in the **L3/G3** and **L4/G4** systems. More detailed structural evidence of the complexes was provided by 2D-COSY and NOESY (Supplementary Figures 86–110). Narrow bands of signals were observed in the 2D diffusion-ordered NMR spectroscopy (2D DOSY) spectra of **G2−G4** (Fig. 3c,d, Supplementary Figure 111a–c), with diffusion coefficient (*D*) values at $2.05 \times 10^{-10}$, $1.65 \times 10^{-10}$, $2.45 \times 10^{-11}$ m$^2$ s$^{-1}$ (log*D* values as −9.69, −9.78, and −10.61), respectively. Due to low solubility of **G4** in CD$_3$CN, **G4** was characterized in DMSO-$d_6$ with high viscosity, and as a result, **G4** showed a lower *D* value compared with **G2** and **G3**. According to the experimental *D* values, the calculated diameters of **G2−G4** was 8.0, 10.2, 11.4 nm, respectively (by using modified Stocks–Einstein equation based on an oblate sphoid model, the calculating method is summarized in SI, Supplementary Figure 134a–c). These results are comparable to

the outer diameter measured from the molecular modeling of **G2**−**G4** (7.9, 9.8, 11.4 nm, respectively in Fig. 1a).

Because of their large sizes, we used atomic force microscopy (AFM), transmission electron microscopy (TEM), and scanning tunneling microscopy (STM) to visualize individual surpamolecules. Both AFM and TEM imaging provided data on nested hexagons' dimensions, including heights and diameters. In AFM imaging, a series of particles with monolayer heights (ca. 2.0 nm) were observed on mica surface for **G2−G4** (Supplementary Figures 121a–d, 122a–d, and 123a–d). TEM imaging on Cu grid also showed uniform nanoparticles with comparable size as molecular modeling (Supplementary Figure 124a–c). In STM (Fig. 4a–f, Supplementary Figure 129a–f), we observed the formation of nanoribbon-like structures through hierarchical self-assembly of nested hexagons on the surface of highly oriented pyrolytic graphite (HOPG), perhaps attributed to the π−π interactions between the conjugated framework of nested hexagons and HOPG, as well as the high affinity of long alkyl chains onto HOPG[41]. The ring-shaped topology observed by STM confirmed the nested structures.

Due to the large and rigid scaffolds, we speculated these nested hexagons might prefer stacking together to hierarchically assemble into tubular-like nanostructures[38]. Indeed, we obtained fiber-like nanostructures through slow diffusion of diisopropyl ether into the DMF solutions of **G2−G4**, confirmed by TEM (Fig. 4g–i, Supplementary Figures 125–127). With short growth time and low concentration, TEM revealed the formation of nanotubes with a uniform internal diameter (ca. 5 nm) and lengths up to several micrometer. The diameter of each nanotube is consistent as the size obtained in molecular modeling given the contribution from alkyl chains, suggesting that the tubules originated from nested hexagons stacking. Increasing the growth time with high concentration, such nanotubes might further assemble together with ABAB pattern through hexagonal close packing[38]. Fiber-like nanostructures were also observed in TEM images of **G2** in DMSO/H$_2$O (v/v, 1/2) mixture (Supplementary Figure 128a–d), although the molecules were mainly aggregated rapidly to form fine packing shapes. Additionally, no hierarchical assembly or aggregation of the nested hexagon molecules was

**Table 1 The antimicrobial activity and selectivity of organic ligands and supramolecules**

|  | G1 | G2 | G3 | G4 | L1 | L2 | L3 | L4 | Daptomycin |
|---|---|---|---|---|---|---|---|---|---|
| MRSA (MIC, µg/mL) | 30 | 3 | 0.5 | 0.5 | >100 | >100 | >100 | >100 | 0.5 |
| *E. coli* (MIC, µg/mL) | >100 | >100 | >100 | >100 | >100 | >100 | >100 | >100 | – |
| Hemolysis (H$_{50}$) | >250 | >250 | >250 | >250 | >250 | >250 | >250 | >250 | – |
| Selectivity (H$_{50}$/MIC) | >8 | >83 | >500 | >500 | – | – | – | – | – |

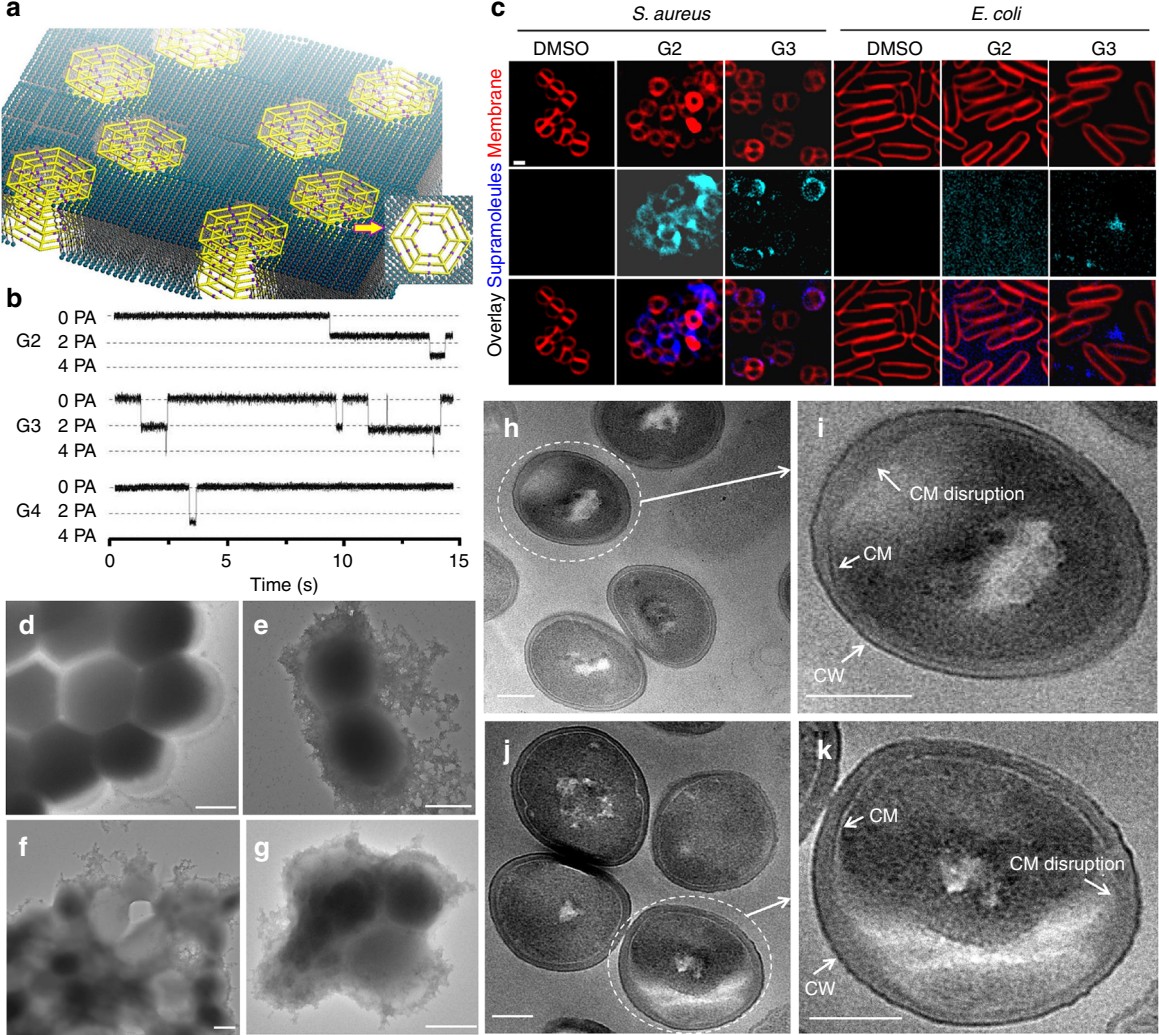

**Fig. 5** Membrane activity and antimicrobial action study of nested hexagons. **a** Proposed model of transmembrane channels formed by **G3**. **b** Current traces (15 s) of **G2** (5.0 nM), **G3** (5.0 nM), and **G4** (40 nM) in the planar lipid bilayer at +100 mV in KCl (1.0 M) solution. **c** 3D deconvolution fluorescence microscopy images of bacteria cells with and without treatment of nested hexagons (4 µM, FM4-64 dye was added) Scale bar: 1 µm. TEM images of MRSA cells (**d**), MRSA cells without treatment. **e**−**g** MRSA cells treated by **G2**: **e** supramolecules aggregated on the cell surface; **f, g** cells were damaged and cell death was observed. Scale bars: 500 nm. Ultrathin sectioning TEM images of MRSA after treatment with **h, i** G2 (25 nM); **j, k** G3 (25 nM). MRSA cells had well-defined cell wall (CW) but disrupted cell membranes (CM). Scale bars: 200 nm

observed in homogeneous solution, i.e., acetonitrile or DMSO based on concentration-depentent $^1$H-NMR spectra of **G2** (Supplementary Figures 112a–e and 113a–d), which do not show any obvious change.

**Antimicrobial activity study**. We expected that nested hexagons might possess antimicrobial activity based on the following speculations: (1) The highly positive charged nested hexagon may have strong electrostatic interaction with the negatively charged anionic glycopolymers, i.e., teichoic acids on the cell envelope[42]; (2) pyridinium polymers are well-known for their membrane-disrupting activity against bacteria[43]; (3) metal complexes have also been reported with antimicrobial activity[33]. Based on this motivation, the antibacterial activity and selectivity of **G1**−**G4** were assessed against Gram-positive bacterium MRSA and Gram-negative bacterium *E. coli* (Table 1). **G2**−**G4** displayed remarkable antimicrobial activity against MRSA as daptomycin, which is the first clinically approved lipopeptide antibiotic against multi-drug-resistant, Gram-positive pathogen MRSA[44]. The minimum

inhibitory concentrations (MICs) against MRSA are 3, 0.5, and 0.5 μg/mL, or 167, 18, and 13 nM for **G2−G4**, respectively. **G4** did not show significant increase in inhibition effect on microbial growth compared to **G3**. This is mainly because **G4** precipitated when in contact with the growth medium. Therefore, increasing the solubility of high generation of nested hexagons may enable more efficient studies in biological media and translate to effective antimicrobial activity. In contrast, **G1** exhibited weak anti-microbial activity, perhaps the multiple entities, i.e., square to hexagon (Supplementary Figure 114), were unable to assemble into tubular structures. In addition, the smaller size of **G1** was expected to lead to weaker interaction with bacterial membranes. Note that pyridinium salts ligands (**L2−L4**) with a few cationic charge(s) did not show comparable activity as reported in pyridinium polymers under the tested conditions[43]. **G1−G4** were unable to inhibit *E. coli* growth possibly owing to the presence of both inner and outer membranes in Gram-negative bacteria[45], which blocked penetration of these supramolecules.

Furthermore, red-blood-cell hemolysis studies displayed negligible hemolytic toxicity of these antimicrobial materials, as the surfaces of Gram-positive bacteria are more negatively charged due to the presence of teichoic acids, whereas the outer leaflet of mammalian cell membrane are largely zwitterionic[46]. Therefore, the electrostatic interaction between the surfaces of the bacteria and the cationic supramolecules is much stronger than that between the surfaces of the red blood cells and the cationic supramolecules, leading to excellent antimicrobial activity and selectivity.

According to the electroporation model of cationic peptides on the surface of bacteria[47], the formation of large supramolecules before coming into contact with the cell surface is expected to enable more efficient interaction with the cell membrane than the individual component, i.e., ligands or metal ions, due to large surface area and high density of cationic charge. Such 'synergistic' effects are of interest as they may help circumvent resistance and reduce the toxicity of conventional antibiotics. Indeed, the resistance of MRSA to cadmium salts can be >1 mg/mL[48], much higher than the MIC of **G2−G4**. We speculated that the toxicity of Cd(II) ions could be substantially reduced in the coordination with multi-armed tpy ligands, which act as the chelation agent like EDTA used in the standard chelation therapy for Cd(II). Therefore, the antibacterial potency of supramolecules should be derived from the nested hexagons rather than individual component. We hypothesized that these nested hexagons could insert into bacterial membrane to form transmembrane channels (Fig. 5a) depending on the charge and hydrophobicity and cause the leakage of cytoplasmic components and cell death[49]. We next performed detailed study to address the mechanism of antimicrobial action.

The membrane activity of the compounds was first investigated by conductance measurements on a planar lipid bilayer composed of the mixture of 1,2-dipalmitoyl-sn-glycero-3-phosphatidylglycerol (DPPG) and 1,2-dipalmitoyl-sn-glycero-3-phosphatidylethanolamine (DPPE) (9:1, molar ratio)[50], the two most abundant phospholipids in bacterial membranes. In the presence of supramolecules, the currents across the bilayers at −100 mV potential showed square ion-conductance signals for all **G2−G4** (Fig. 5b). These results suggested that these three nested hexagons could incorporate into the lipid bilayer and form transmembrane channels. More importantly, the conductance of channel formed by **G2−G4** was calculated to be 15, 21, and 26 pS, respectively.

Second, subcellular localization of **G2−G4** was monitored using 3D deconvolution fluorescence microscopy (Fig. 5c), which is a combination of optical and computational techniques to maximize the observed resolution and signal from a biological specimen[51]. Note that due to phototoxicity in imaging live cells, it

is challenging to use confocal laser scanning microscopy (CLSM) to image the membrane activity. In the test, bacterial cells, i.e., *S. aureus* and *E. coli* were incubated with supramolecules for 5 min at room temperature. High concentration (4 μM) of nested hexagons was applied to ensure the strong fluorescence for detection. FM4-64 dye was added to visualize cell membrane immediately prior to imaging. Compared to FM4-64 with emission at 640 nm, **G2** and **G3** have distinct fluorescent emission maximum at 510 and 525 nm (Supplementary Figure 133a–d), respectively, while **G4** emits yellow light (maximum at 550 nm) close to the emission wavelength of FM4-64 (Supplementary Figure 133e, f). Fluorescence of FM4-64 and supramolecules **G2** and **G3** were captured with TRITC and DAPI filters, respectively. As shown in Fig. 5c, **G2−G3** showed fluorescence on the membrane of Gram-positive *S. aureus*; whereas no membrane-associated fluorescence was detected for *E. coli*, indicating that nested hexagons could not interact with the Gram-negative bacterium. We also conducted another fluorescence study without using membrane dye FM4-64 for staining with supramolecules **G2**, **G3**, and **G4** (Supplementary Figure 132a, b). Interestingly, in the absence of the membrane stain all nested hexagons could insert into *S. aureus* membrane and be imaged even at lower concentration (2 μM/mL). We speculated that there could be a competition between membrane dye molecules and supramolecules in the interaction with *S. aureus* membrane. As such, fluorescence microscopy results confirm that these nested hexagons interacted with *S. aureus* but not *E. coli*. membrane.

Third, TEM was used explore the mechanism of the antimicrobial action of **G2−G4** on the morphological changes in MRSA before and after incubation with supramolecules for 8 h at lethal doses (25 nM). As shown in Fig. 5d–g, the cell envelopes of the microorganisms were damaged, and cell lysis was observed after treatment with supramolecule **G2**. In addition, a large empty space was observed in the cytosol, as well as leaking of what appeared to be cytoplasmic contents (Fig. 5f, g). We reasoned that the cationic supramolecules can be easily adsorbed by the negatively charged anionic glycopolymers, i.e., teichoic acids[42] on the cell wall of MRSA, pass through the periplasmic space, then stack into channels in the inner lipid membrane due to the strong intermolecular interactions, e.g., π−π interaction of backbone and Van der Waals interactions of alkyl chains, as well as the hydrophobic interactions between supramolecules and lipid layers. Similar morphological changes were observed by incubating **G3** and **G4** with MRSA (Supplementary Figures 130 and 131). To further determine the location of disruption, we performed an ultrathin sectioning TEM study[50] of MRSA cells after treatment with **G2** and **G3** (Fig. 5h–k). TEM analysis indicated that the structure of cell wall was maintained, whereas the morphology of the bilayer underwent profound changes and membrane integrity was compromised in response to drug treatment. This result highly supported the hypothesis that the antimicrobial activities of **G2−G4** were primarily mediated by the membrane interaction and disruption of the bacterial membrane by forming transmembrane channels.

## Discussion

In the seeking of efficient supramolecular expression of nested structures, we designed and assembled three generations of 2D multilayered concentric supramolecular architectures through coordination-driven self-assembly. In the ligand synthesis, we introduced Suzuki coupling reaction on the pyrylium salts followed by ring-closing and consecutive condensation reaction between pyrylium salts and precursors with primary amines to significantly simplify the preparation of multitopic building blocks. After self-assembly, these giant 2D supramolecules (**G2**

−G4) have strong tendency towards forming tubular nanostructures through hierarchical self-assembly. Such structural features based on the desired shapes and sizes, cationic scaffolds with precise charge positioning, balance of the overall hydrophobicity/hydrophilicity, and remarkable rigidity and stability enabled their application as antimicrobial agents. G2−G4 displayed high antimicrobial activity against Gram-positive bacteria MRSA and negligible toxicity to eukaryotic cells. The membrane activity of these supramolecules was confirmed via electrophysiology study of planar lipid bilayer, subcellular localization by 3D deconvolution fluorescence microscopy, and bacterial morphology by TEM. We believe that those 2D multilayered supramolecules were able to assemble into channels with multilayered structure and distinct pore size inside the bacterial membrane, and thus lead to enhanced leakage of cytoplasmic components and cell death. Our endeavors will shed light into both antibiotics and supramolecular chemistry field and pave a new avenue into the development and application of antimicrobial agents.

**Data availability**. The data that support the findings of this study are available from the authors on reasonable request, see author contributions for specific data sets.

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

## Acknowledgements

This research was supported by the National Science Foundation (CHE-1506722 to X.L.; CHE-1351265 to J.C.), National Institutes of Health (1R01GM128037-01 to X.L.), startup grant from the University of South Florida (P.E.), National Natural Science Foundation of China (No. 21206016 to W.G.; 21422202 to J.-L.H.; 21528201 to X.L. and X.-Q.H.), the Fundamental Research Funds for the Central Universities (DUT17LK07 to W.G.), and the Program for Science & Technology Innovation Talents in Universities of Henan Province (17HASTIT004 to X.-Q.H.).

## Author contributions

X.L. conceived and designed the experiments. H.W., X.Q., M.W., X.G., and X.-Q.H. completed the synthesis. H.W., X.Q., X.J., M.W., and Y.L. conducted NMR, MS, and TEM characterization. K.W. and B.X. performed STM. M.S. and J.C. designed and performed the antimicrobial and toxicity studies. R.B. and P.E. performed fluorescence microscopy imaging. W.-W.H. and J.-L.H. performed planar lipid bilayer and ultrathin sectioning TEM study. H.W., W. G., J.-L.H., P.E., J.C., and X.L., analyzed the data and wrote the manuscript. All the authors discussed the results and commented on and proofread the manuscript.

## Additional information

**Competing interests:** The authors declare no competing financial interests.

