## [Peer Review File · Nature Communications]

Reviewers' comments:

Reviewer #1 (Remarks to the Author):

This is a lovely paper by Xiaopeng Li and co-workers that describes the synthesis and antibacterial properties of self-assembled nested supramolecular hexagons. I have always found the structures prepared by Li to be stunning and this paper is no exception. What is even more impressive is that Li has begun to demonstrate the functions of his supramolecules. In this paper he shows that the nested hexagons display high activity against methicillin resistant *s. aureus*. The results will be of great interest to all supramolecular chemists who are always looking for real world applications of their structures but also to the broader chemical community. I am happy to recommend publication after minor revisions.

- 1) Abstract - the authors attribute the antimicrobial activity to several factors (precise charge positioning, hydrophobicity / hydrophilic balance, rigidity / stability) but from my point of view the author really just know that G2-G4 work and the building blocks don't. One should not make such speculative comments on the sources of activity without extensive structure activity relationships.
- 2) Page 7 - the discrepancy between DOSY derived diameter and molecular modeling is rather large. Might this be due to the disc like shape of G2-G4 not being so appropriate for the model used to go from diffusion coefficient to diameter.
- 3) Figure 1 - panel c is really quite crowded / messy looking / hard to follow. Couldn't structures 6, 9, 17, 12, 19 be combined into one structure with different R- groups in the para position and different values of 'n' for the phenylene spacers. That might allow the figure to be a little clearer. Also, the font sized for the part labels e.g. a) b) are very large and the atom labels are very small. Try to be more consistent.
- 4) Page 9 - the authors do not address whether the stacking of the hexagons into tubes also occurs in homogenous solution. The NMRs look clean so maybe not, but perhaps solvent is critical in this regard. I suggest the authors comment on this.

Reviewer #2 (Remarks to the Author):

This is an important paper presented Dr. Li and coworkers about the development of novel nested supramolecular hexagons and the study of their antibacterial activity. The authors demonstrate that: 1) upon properly designing the pyridyl ligands, well-defined nested supramolecular hexagons were obtained via coordination-driven self-assembly; 2) These hexagons can further self-assemble into large nanotubes; 3). These structures show promising antibacterial activity. The authors have done an excellent job on characterizing the supramolecular structures using 1D and 2D NMR, ESI-MS, AFM, and TEM. The proposed mechanism of action of their antibacterial activity is interesting. All these findings are novel within the field of metal-based supramolecular chemistry. This work will attract great attentions from the scientific community of supramolecular chemistry and others. Therefore, I recommend the acceptance of this paper. Below are a few minor suggestions to improve the manuscript:

1. All the MS of the newly synthesized ligands should be high resolution MS with at least 4 digits past the decimal point.
2. It would be valuable to see the hierarchical self-assembly of the hexagons in water instead of DMS, since these structures are being used in aqueous solution for antibacterial study.

Response Letter to Reviewers' Comments

Reviewer 1

1) Abstract - the authors attribute the antimicrobial activity to several factors (precise charge positioning, hydrophobicity / hydrophilic balance, rigidity / stability) but from my point of view the author really just know that G2-G4 work and the building blocks don't. One should not make such speculative comments on the sources of activity without extensive structure activity relationships.

Response: The last sentence in the abstract was removed.

2) Page 7 - the discrepancy between DOSY derived diameter and molecular modeling is rather large. Might this be due to the disc like shape of G2-G4 not being so appropriate for the model used to go from diffusion coefficient to diameter.

Response: We made mistake in the description of the size obtained from DOSY. The values calculated from DOSY should be "radii" instead of "diameters". Note that our ring-shaped like shape of supramolecules are not appropriate to use the original Stocks-Einstein equation based on sphere model. As a result, according to some similar reports (*Angew. Chem. Int. Ed.* **47**, 2008, 2235-2239 and *Org. Biomol. Chem.* **12**, 2014, 7932-7936), we used the modified equation based on oblate spheroid model. The detailed calculation procedure has been summarized in SI, and the results are fitting well with the modeling results. In order to more easily compare the experimental and modeling results, we converted the value derived from diffusion coefficient from radii to diameter by doubling the original numbers.

3) Figure 1 - panel c is really quite crowded / messy looking / hard to follow. Couldn't structures 6, 9, 17, 12, 19 be combined into one structure with different R- groups in the para position and different values of 'n' for the phenylene spacers. That might allow the figure to be a little clearer. Also, the font sized for the part labels e.g. a) b) are very large and the atom labels are very small. Try to be more consistent.

Response: Figure 1 has been modified according to the suggestion.

4) Page 9 - the authors do not address whether the stacking of the hexagons into tubes also occurs in homogenous solution. The NMRs look clean so maybe not, but perhaps solvent is critical in this regard. I suggest the authors comment on this.

Response: We performed concentration dependent ¹H-NMR experiments of G2 in two good solvents, i.e., CD₃CN (from 10.0 mg/mL to 0.6 mg/mL) and d₆-DMSO (from 5.0 mg/mL to 0.6 mg/mL). No significant changes were observed by varying concentration, suggesting that no aggregation happened in the homogenous solution, as shown in Figures R1 and R2. The corresponding discussion was added into Page 10 with high light. And the concentration dependent ¹H-NMR spectra are added in SI Figures S112 and S113.

Figure R1. Concentration dependent ^1H NMR (400 MHz, CD_3CN , 300 K) spectra of complex **G2**, (a) 0.6 mg/mL, (b) 1.2 mg/mL, (c) 2.5 mg/mL, (d) 5.0 mg/mL, (e) 10.0 mg/mL.

Figure R2. Concentration dependent ^1H NMR (400 MHz, $\text{DMSO-}d_6$, 300 K) spectra of complex **G2**, (a) 0.6 mg/mL, (b) 1.2 mg/mL, (c) 2.5 mg/mL, (d) 5.0 mg/mL.

Reviewer 2

1. All the MS of the newly synthesized ligands should be high resolution MS with at least 4 digits past the decimal point.

Response: All the high resolution MS data have been collected. The data have been summarized in the corresponding ligand syntheses section in SI, and the spectra have been added in Figs S79-83.

2. It would be valuable to see the hierarchical self-assembly of the hexagons in water instead of DMSO, since these structures are being used in aqueous solution for antibacterial study.

Response: According to the reviewer's suggestion, we chose **G2** as the model system and collected its TEM images in DMSO/H₂O (v/v, 1/2) mixture. Fiber-like nanostructures were also observed in the mixture, although the molecules were mainly aggregated rapidly to form fine packing shapes, as shown in Figure R3. The corresponding discussion was added into Page 10 with high light. And the TEM images are concluded in SI Figure S128.

Figure R3. TEM imaging of aggregation of **G2** in DMSO/H₂O (v/v 1/2, 0.5 mg/mL).

REVIEWERS' COMMENTS:

Reviewer #1 (Remarks to the Author):

I have looked over the revisions made by the authors in response to the previous reviews. I believe they have responded fully and that the manuscript is now ready for publication.

Reviewer #2 (Remarks to the Author):

The authors have responded to my comments properly, and I recommend the acceptance of this manuscript.